# Quaternized Polysulfone as a Solid Polymer Electrolyte Membrane with High Ionic Conductivity for All-Solid-State Zn-Air Batteries

**DOI:** 10.3390/membranes15040102

**Published:** 2025-04-01

**Authors:** Luis Javier Salazar-Gastélum, Alejandro Arredondo-Espínola, Sergio Pérez-Sicairos, Lorena Álvarez-Contreras, Noé Arjona, Minerva Guerra-Balcázar

**Affiliations:** 1Facultad de Ingeniería, División de Investigación y Posgrado, Universidad Autónoma de Querétaro, Querétaro 76010, Santiago de Querétaro, Mexico; luis.javier.salazar@uaq.mx; 2Centro de Investigación y Desarrollo Tecnológico en Electroquímica S. C., Sanfandila 76703, Pedro Escobedo, Mexico; jarredondo@cideteq.mx; 3Centro de Graduados e Investigación en Química, Instituto Tecnológico de Tijuana, Tecnológico Nacional de México, Blvd. Alberto Limón Padilla, S/N Col. Otay Tecnológico, Tijuana 22510, Baja California, Mexico; sperez@tectijuana.mx; 4Centro de Investigación en Materiales Avanzados S. C., Complejo Industrial Chihuahua, Chihuahua 31136, Chihuahua, Mexico; lorena.alvarez@cimav.edu.mx

**Keywords:** Zn-air batteries, separator, solid polymer electrolyte, all-solid-state

## Abstract

Solid polymer electrolytes (SPEs) are gaining attention as viable alternatives to traditional aqueous electrolytes in zinc–air batteries (ZABs), owing to their enhanced performance and stability. In this study, anion-exchange solid polymer electrolytes (A-SPEs) were synthesized via electrophilic aromatic substitution and substitution reactions. Thin films were prepared using the solvent casting method and characterized using proton nuclear magnetic resonance (¹H-NMR), Fourier-transform infrared spectroscopy (FT-IR), and thermogravimetric analysis (TGA). The ion-exchange capacity (IEC), KOH uptake, ionic conductivity, and battery performance were also obtained by varying the degree of functionalization of the A-SPEs (30 and 120%, denoted as PSf30/PSf120, respectively). The IEC analysis revealed that PSf120 exhibited a higher quantity of functional groups, enhancing its hydroxide conductivity, which reached a value of 22.19 mS cm^−1^. In addition, PSf120 demonstrated a higher power density (70 vs. 50 mW cm^−2^) and rechargeability than benchmarked Fumapem FAA-3-50 A-SPE. Postmortem analysis further confirmed the lower formation of ZnO for PSf120, indicating the improved stability and reduced passivation of the zinc electrode. Therefore, this type of A-SPE could improve the performance and rechargeability of all-solid-state ZABs.

## 1. Introduction

The need for renewable energy sources such as solar and wind energy is critical because of the rapid development of the global economy [1,2]. Efficient utilization of these clean energy sources has driven significant interest in various electrochemical energy storage and conversion technologies designed to meet diverse energy needs [3]. Among these technologies, zinc–air batteries (ZABs) have emerged as a particularly promising option in the field of energy storage owing to their high theoretical energy density, low cost, and inherent safety [4,5].

The electrolyte, which has traditionally been in liquid form (6 M KOH), is a critical component for efficient and safe performance of Zn–air batteries. However, liquid electrolytes present significant challenges such as volatility, leakage, and limited long-term stability [6]. These issues compromise the safety and environmental sustainability of batteries and limit their application in portable devices. To address these challenges, transition to solid-state electrolytes has gained attention as a viable alternative [7]. Solid-state electrolytes not only eliminate issues associated with liquid electrolyte leakage and evaporation but also enhance the overall safety and environmental sustainability of batteries [8,9]. Moreover, solid-state electrolytes can function as both ionic conductors and physical separators, preventing internal short circuits and simplifying the design and fabrication of batteries [10].

Anion-exchange membranes (AEMs) have been extensively explored for diverse electrochemical applications because of their versatility. For instance, AEMs based on quaternary ammonium-functionalized polymers are pivotal in fuel cells because of their hydroxide ion conductivity under alkaline conditions [11]. In electrodialysis desalination, tailored AEMs enable efficient ion separation [12], whereas composite AEMs with enhanced mechanical strength are critical for water electrolyzers [13]. A recent study by Wang et al. demonstrated nanostructured AEMs with hierarchical porosity for high-performance energy devices, emphasizing the role of the membrane architecture in ion transport optimization [14]. Similarly, hybrid AEMs integrated with catalytic layers have improved oxygen reduction kinetics in aluminum–air batteries [15], underscoring the importance of material design across applications. Recent advances in anion-exchange membranes (AEMs) have highlighted the effectiveness of cross-linking strategies in enhancing performance. Wei et al. [16] reported a GA cross-linked AEM with a conductivity of 0.024 S cm^−1^ and low swelling, reinforcing the potential of chemical cross-linking to optimize both the conductivity and mechanical integrity in solid electrolytes. These studies highlight the critical role of tailored membrane designs in advancing high-performance AEMs for electrochemical applications.

In this context, A-SPEs have emerged as a promising solution [17]. Among the various available polymer materials, quaternized polysulfone (QPSf) is a particularly interesting candidate for use in solid polymer electrolytes. Polysulfones (PSf) are known for their excellent thermal and chemical stability, as well as their good mechanical properties [18]. The functionalization of PSf through quaternization introduces anionic groups that can facilitate ion conduction, making it suitable for application in anion-exchange batteries. Furthermore, these solid electrolytes can prevent the migration of zinc ions and formation of carbonates during battery operation, thus improving battery durability. However, the water-retaining properties of traditional membranes are relatively poor and directly limit the discharge of flexible, rechargeable ZABs [19,20].

Despite these advantages, the use of QPSfs as solid polymer electrolytes in zinc–air batteries has rarely been studied. Recent studies have predominantly focused on other types of polymers and electrolyte systems, leaving a significant gap in the literature regarding the potential of QPSfs for this application. These types of polymers have been used in other electrochemical technologies, such as fuel cells, electrodialysis desalination, water electrolyzers, and aluminum–air batteries [21,22,23,24,25]. Regarding the few studies found for ZABs, Zhang et al. reported the development of a solid-state polyelectrolyte for use in flexible Zn–air batteries. The solid electrolyte was synthesized through chemical functionalization of cellulose fibers and graphene oxide, followed by layer-by-layer filtration, crosslinking, and ion exchange processes. This solid electrolyte was evaluated in a rechargeable Zn–air battery and compared to the anion exchange membrane A201 from Tokuyama Corporation [20]. The nanocomposite solid polyelectrolyte exhibited superior performance in terms of the current density compared to the A201 membrane [26]. Similarly, Xu et al. developed a system using cellulose nanofiber matrices confined with ionic liquids to create flexible electrolytes with high ionic conductivity. This system achieved high hydroxide ion conductivity owing to the high hydrophilicity of the cellulose nanofibers and ionic liquids, which provided a solid polyelectrolyte with numerous conductive channels [27]. Alkaline membranes play a dual role as charge carriers and electrode separators in electrochemical systems, critically influencing the performance and durability. Their importance has surged in energy technologies, particularly in fuel cells, for which extensive research has been conducted. However, their widespread adoption remains limited by two key challenges: conductivity and stability. Strategies such as engineering microphase-separated structures (to accelerate ion transport) and optimizing micropore networks (to reduce resistance) have proven to be effective. These innovations have driven conductivity from <10 mS cm^−1^ to >100 mS cm^−1^, marking transformative progress in the field [28].

The introduction of QPSFs into the design of solid polymer electrolytes could be beneficial for zinc–air batteries by providing safer, more efficient, and more durable all-solid-state ZABs. This study aims to explore the performance of QPSF-based solid polymer electrolytes in all-solid-state ZABs. For this purpose, different degrees of functionalization of the QPSFs were investigated to evaluate their impact on ionic conductivity and electrochemical stability. The battery performance (power density and charge/discharge behavior) of the resulting A-SPEs was compared with that of the benchmark Fumapem FAA-3-50 alkaline membrane (FuelCellStore ^®^), selected because of its status as one of the most extensively studied anion-exchange membranes for alkaline systems and its structural similarity to QPSF-based electrolytes (both utilize quaternary ammonium cations as ion-conductive groups).

## 2. Materials and Methods

### 2.1. Reagents

Polysulfone (Mw = 35,000), chloroform (≥99.5%), triethylamine (≥99%), *N*-methyl-2-pyrrrolidone (≥99%), paraformaldehyde (95.0%), chlorotrimethylsilane (≥98.0%), tin chloride (98.0%), ethyl acetate (≥99.5%), diethylether (≥99.7), dimethylsofoxide-d6 (99.9% D), chloroform-d1 (99.8% D) were purchased from Sigma-Aldrich (Saint Louis, MO, USA). Potassium hydroxide (85.0%) was purchased from Merck (Darmstadt, Germany). Hydrochloric acid, ethanol (95.0%) was purchased from J.T. Baker (Mexico City, Mexico). Bromothymol blue (95.0%) was purchased from Meyer (Mexico City, Mexico). All chemicals were used without further purification.

### 2.2. Chloromethylation and Quaternization of Polysulfone

To obtain chloromethylated polysulfone (PSf−Cl), an electrophilic substance was used following a previously reported methodology to avoid the use of chloromethyl methyl ether, which is dangerous for human health [29]. In a round-bottom flask, PSf pellets (2 g) were dissolved in chloroform under vigorous magnetic stirring. Following the addition of 44.96 mmol of paraformaldehyde, 0.44 mmol of tin chloride (IV) was added to the suspension dropwise. Finally, 44.96 mmol of chlorotrimethylsilane was added dropwise using the procedure described by Avram et al. [30]. Different functionalization degrees were achieved during the chloromethylation reaction by adjusting the reaction duration. After the reaction was complete, the solution was stirred and precipitated using cold methanol. After filtration and multiple methanol washes, the precipitated PSf functionalized with CH_2_Cl groups (PSf−Cl) was dried in a vacuum oven at 50 °C. The quaternization reaction involved using those precursors with different degrees of functionalization; 1.5 g of PSf−Cl was dissolved into N-methyl-pyrrolidone (NMP) using magnetic stirring in a round-bottom flask, followed by the addition of 1 mL of triethylamine (TEA), then the mixture was heated up for 24 h at 80 °C to replace Cl with amine groups. The mixture was then precipitated with diethyl ether and washed with 99.5% ethyl acetate. Brown powder (QPSf) was obtained and kept at 60 °C in a vacuum oven [31].

### 2.3. Preparation of Alkaline Solid Polymer Electrolyte

A-SPEs were prepared using the solvent casting method, as described below. A certain amount of polymer powder was dissolved in NMP to obtain a polymer solution with a concentration of 16 wt. %. The polymer solution was cast onto a glass petri dish. The solvent was removed at 50 °C under ambient pressure for 48 h, and an A-SPE was formed, which was subsequently peeled off from the petri dish via immersion in DI water. A-SPE was washed with DI water several times to remove residual NMP. Subsequently, A-SPE in Cl^−^ form was obtained and stored in DI water before use. To obtain the OH^−^ form, A-SPE in the Cl^−^ form was immersed in 1 M KOH at room temperature for 48 h. 

### 2.4. Characterization

^1^H NMR spectra of the synthesized polymers were recorded at 400 MHz with a Bruker Advance III HD spectrometer at 30 °C. Chemical shifts were referenced to the peak of tetramethyl silane (TMS). The polymers were dissolved in chloroform-d for chloromethylated polysulfone and DMSO-d6 for quaternized polysulfone. FT-IR spectra of the A-SPEs were recorded using a Perkin Elmer Spectrum Gx with attenuated reflectance (ATR) in a scanning range of 600–4000 cm^−1^, with a resolution of 2 cm^−1^ and 16 scans. Thermogravimetric analysis (TGA) was performed under nitrogen atmosphere using a TQ500 instrument. Thermogravimetric curves were recorded in the temperature range of 30 °C to 800 °C at a heating rate of 20 °C min^−1^. To study membrane morphology, SEM images were acquired using a microscope (JEOL, model JSM-6610LV, Akishima, Japan) operated at an accelerating voltage of 15 kV.

The KOH uptake for the A-SPEs in the OH^−^ form was calculated based on the changes in the mass of the swollen and dried states. Each A-SPE was kept in a 1 M KOH aqueous solution for 24 h, excess of KOH solution was removed with tissue paper, and the weight of the wet membrane was recorded. The A-SPEs were dried in an oven at 60 °C until a constant weight was obtained. The KOU was calculated using Equation (1):(1)$$KOH\;uptake\left(\%\right)=\frac{m_{wet}-m_{dry}}{m_{dry}}*100$$
where m_wet_ and m_dry_ are the weights of the swollen and dried membranes, respectively.

The ion exchange capacity (IEC) was measured using an acid–base titration method as previously reported [32]. The A-SPEs were cut into pieces of 2 cm × 2 cm, then they were immersed in 15 mL of 0.01 M HCl and kept for 24 h to ensure the consumption of OH^−^ with H^+^. Then, the remaining acidic solution was transferred to a flask and titrated with 0.01 M KOH using bromothymol blue (1%) as an indicator. The IEC was recorded in meq of H^+^ g^−1^ and was determined based on the difference in the number of milliequivalents of HCl solution before (meq of HCl)_0_ and after (meq of HCl)_f_ the membrane being neutralized, according to Equation (2):(2)$$IEC\left(mmol\;g^{-1}\right)=\frac{\left(V_{b}-V_{s}\right)*C_{HCl}*1000}{m_{dry}}$$
where V_b_ and V_s_ are the consumed volumes (L) of the KOH solution for the blank sample and A-SPE sample, respectively, C_HCl_ is the concentration (mol L^−1^) of the HCl solution, and m_dry_ is the mass of the dry A-SPE.

The ionic conductivities (σ) of the A-SPEs were measured using a four-probe method with a VSP-3 Potentiostat/Galvanostat over a frequency range of 0.1 to 500 kHz in a Teflon conductivity cell (BT-112, Scribner, Southern Pines, NC, USA). A-SPE samples in the OH^−^ form were cut into 1 × 2.5 cm rectangular strips. The SPE strip was placed between the inner platinum wires to ensure contact with both the outer platinum wires and platinum mesh. A clamp was then applied to secure the membrane surfaces and was tightened using screws. The conductivity cell was then placed in a water bath at room temperature (25 °C). The resistance of the A-SPE was determined using Nyquist plots. The hydroxide conductivity was calculated using Equation (3):(3)$$\sigma (mS\;{cm}^{-1})=\frac{L}{RWT}$$
where σ represents the hydroxide conductivity of the SPE, L is the distance between the inner platinum wires (5.5 cm), R is the resistance of the SPE (obtained from the high-frequency intercept on the real impedance axis), and W and T are the width and thickness of the SPE in its wet form, respectively [33,34].

### 2.5. Zn–Air Battery Test

For the battery tests, a design consisting of two acrylic slotted plates that were mechanically sealed with screws was used. This design allowed tests to be conducted using small amounts of material, and the contact area between the cathode and anode was 0.06 cm^2^. A Sigracet 33 B (FuelCellStore ^®^) cathode was used as the current collector and coated with Pt-IrO_2_/C. The metallic charge used in the air electrode was 1 mg cm^−2^, and a piece of polished metallic zinc (99.9% purity) was used as the zinc electrode. In all cases, the membranes were activated in 1 M KOH for 24 h and then immersed in 6 M KOH before the cell assembly. Electrochemical Impedance Spectroscopy from 100 KHz to 100 Hz was used to evaluate the correct assembly of the ZAB. Linear voltammetry was used to evaluate the battery performance of different A-SPE materials. Charge and discharge cycles were evaluated at −0.5 mA cm^−2^ at discharge and 0.5 mA cm^−2^ at charge. Figure 1 illustrates the battery assembly process.

## 3. Results and Discussion

Figure 1 shows the ^1^H NMR spectra of PSf (a) and PSf−Cl (b). The PSf spectrum showed peaks corresponding to isopropyl groups of the main chain and phenyl groups at 1.69 ppm and 7–8 ppm, respectively, where hydrogens close to the sulfone group appeared at 7.9 ppm. The ^1^H NMR spectrum of PSf−Cl shows a new peak at 4.56 ppm (*H_f_*), corresponding to hydrogens arising from the incorporation of the chloromethyl group, which add a new chemical environment to the aromatic hydrogen in the ortho position of the isopropyl group. The amount of CH_2_Cl incorporated into the backbone of PSf was calculated from the ^1^H NMR spectra of PSf−Cl according to Pérez-Prior [35] (Equation (4)):(4)$$DC=\frac{\int \frac{H_{f}}{2}}{\int \frac{H_{e}}{6}}*100$$
where ∫H_f_ and ∫H_e_ are the integral area of the signal at 4.56 ppm and 1.69 ppm, respectively. The DC value indicates the degree of chloromethylation.

Figure 1c shows the ^1^H NMR spectrum of quaternized PSf, where the signal corresponding to the methylene of the chloromethyl group shifts to lower ppm values. It was also observed that the signal split into two peaks because the reaction did not occur completely, indicating the presence of residual chloromethyl groups in the quaternized polymer. Two quaternized polymers were prepared: PSf30, obtained from PSf−Cl with a degree of functionalization (DC) of 30, and PSf120, obtained from PSf−Cl with a DC of 120%. It is important to point out that at 3.2 ppm, a new signal appeared that corresponds to methylene in triethylamine, which was linked due to the substitution reaction. In addition, the signal of the methyl groups of the amino compound appeared at a chemical shift of 1 ppm, and the spectrum of the quaternized polymer was obtained in DMSO-d6. This explains the peak at 3.4 ppm, attributed to the solvent and the peak at 2.54 ppm, related to the residual DMSO [36,37].

Figure 2 shows the FT-IR spectra of PSf and PSf−Cl, where the peak at 2968 cm^−1^ corresponds to the aliphatic stretching of C–H from the isopropyl groups of the backbone of PSf, and the peak at 1150 cm^−1^ corresponds to the symmetric stretching of the sulfone group. Asymmetric stretching of the sulfone group was observed at 1320 cm^−1^ and 1296 cm^−1^ [38]. Finally, other signals were observed at 1238 and 1012 cm^−1^, which were attributed to the asymmetric stretching of the ether species (C–O–C) [39,40]. The change in the shape of the band located at 660 cm^−1^ for the PSf−Cl spectra compared to PSf was attributed to C–Cl, as reported by Yang et al [41]. This is consistent with the ^1^H NMR results, indicating that PSf was modified.

For PSf120 and PSf30, the presence of ammonium quaternary cations was confirmed using FT-IR based on the appearance of three bands: the first two bands, one at 3390 cm^−1^, attributed to the absorption of water from the atmosphere, corresponding to the hydrophilicity of the polymer; one at 1690 cm^−1^ for the moiety of the ammonium cations; and a third at 1010 cm^−1^ due to C–N [42].

TGA was performed to study the thermal stability and confirm the modification of pristine PSf (Figure 3). The figure shows that PSf shows only one stage of decomposition at 550 °C, which is due to the decomposition of the skeleton chain of the polymer. On the other hand, PSf−Cl shows two stages of decomposition: (i) at 350 °C, due to the loss of mass of the pedant groups of CH_2_Cl for the chloromethylation reaction, and (ii) at 505 °C, due to decomposition of the main skeleton of PSf. This value shifted to lower temperatures as the pristine polymer became thermally unstable. For the ammonium-functionalized PSf, the first stage of decomposition occurred at 198 °C for PSf120 and 208 °C for PSf30. This behavior is usually observed for ammonium cations because the temperature of decomposition is reported to be between 180 and 220 °C [35,43]. Both polymers became unstable compared to pristine PSf. Furthermore, the degree of functionalization corresponds to the first stage of decomposition of ammonium-functionalized A-SPEs because PSf120 exhibits a higher mass loss than PSf30.

Table 1 lists the values obtained for IEC and KOH uptake, where the degree of functionalization plays an important role. On one hand, the IEC should be higher when the degree of functionalization increases; that is, when the availability of ammonium cations is higher. This agrees with the IR results, where it can be observed that the band at 1690 cm^−1^ is higher for PSf120. However, the KOH uptake value was higher for PSf120, which was also expected because its OH absorption band was higher. Under similar IEC experimental conditions, A-SPE PSf120 exhibited values similar to those of commercial materials. For hydroxide conductivity values, A-SPE PSf30 showed similar performance to that of Fumapem, indicating that both A-SPEs show a similar mechanism for hydroxide ion mobility in solid polyelectrolytes. 

Although commercial A-SPE exhibits a higher IEC value in terms of ionic conductivity, A-SPE PSf30 shows better performance than commercial A-SPE, which can be attributed to the better dispersion of ammonium moieties on the surface. Moreover, A-SPE PSf120 was the separator with the highest σ value, which may be due to the large quantity of ammonium cation groups compared to PSf30. On the other hand, comparing PSf12 and commercial A-SPE, PSf120 showed a greater capability to attach KOH molecules than commercial A-SPE, indicating that A-SPE is more hydrophilic. These results reveal that this type of polysulfone A-SPE is a promising candidate for use as a membrane and separator in Zn–air batteries.

Optical microscopy images were obtained to analyze the structure of the A-SPEs after activation in 6M KOH (Figure 4). The results showed that both Fumapem and PSF120 exhibited microporous structural characteristics of the AEMs. However, PSF130 displayed a significant reduction in porosity, and its surface primarily featured morphological defects.

Nyquist plots (Figure 5a) revealed an increase in internal resistance in the ZABs when using the developed A-SPEs, primarily attributed to challenges in ion transverse transport across the membrane. Despite minor shifts toward higher resistances for the A-SPEs developed in this study, the measured battery resistance values indicated favorable interfacial interactions between the electrolyte and electrodes, which is a critical factor for functional device performance. Furthermore, the stable resistance values suggest successful cell assembly, achieved without precision tools, such as a torque wrench, highlighting the practicality of the proposed fabrication method. Notably, PSf120, with its high degree of functionalization, demonstrated a resistance value comparable to that of commercial A-SPE, underscoring its potential for scalable applications. This was further confirmed through cathodic linear polarization tests (Figure 5b), which showed poor performance for PSf30 and issues related to mixed control zones (between the activation and ohmic regions) for PSf120. Due to the higher ionic conductivity and KOH uptake values of PSf120, a current density of 74.58 mA cm^−2^ was achieved, representing a 42.43% increase compared to the commercial A-SPE.

Anodic linear polarization tests (Figure 5c) demonstrated the strong performance of PSF120, achieving a current density of 10 mA cm^−2^ at an overpotential of 205.5 mV and 248.5 mV for PSF120 and FUMAPEM, respectively, highlighting its excellent capability for zinc nucleation and redeposition processes at lower overpotentials. Charge/discharge cycles (Figure 5d) showed initial round-trip energy efficiencies of 95.24% and 91.39% for PSf120 and the commercial A-SPE, respectively. By the end of the test, both A-SPEs showed a decrease in round-trip efficiency, reaching 87.38% for PSf120 and 85.72% for the commercial A-SPE. This, combined with a 14% increase in cyclability, demonstrates that PSf120 exhibits greater stability and superior performance compared to the commercial Fumapem A-SPE.

The postmortem analysis of the air electrodes (Figure 6) revealed that both commercial A-SPE (Figure 6a) and PSf120 (Figure 6b) exhibit the formation of micrometric deposits. Through EDS analysis and elemental mapping of the commercial A-SPE (Figure 6c), it was confirmed that the composition of these deposits was associated with carbonate formation (due to the interaction of CO_2_ with adsorbed KOH) and zinc migration [44]. Although similar effects were observed for PSf120, the migration of zinc was less pronounced, likely due to the smaller pore size of this A-SPE. Additionally, a greater formation of carbonates was identified, which was attributed to higher electrolyte absorption [45].

As a result of using porous separators, dendritic growth was observed on the zinc electrodes in both commercial A-SPE (Figure 7a) and PSf120 (Figure 7b) [46]. Owing to the higher cyclability of PSF120, it exhibited larger dendrites, which did not pose a significant risk of sudden battery failure. Elemental analysis of the commercial A-SPE (Figure 7c) revealed the formation of carbonates and ZnO, with zinc electrode passivation identified as the primary cause of the system failure. In contrast, the use of PSf120 (Figure 7d) resulted in a higher presence of metallic zinc on the electrode surface and the formation of carbonates. This suggests that PSf120 enhances the electrochemical reversibility of zinc compared to commercial A-SPE, likely due to a higher distribution of atomic oxygen, indicating that this oxygen is associated with ZnO [47].

## 4. Conclusions

Anion-exchange solid polymer electrolytes were obtained by modifying polysulfone with ammonium cations, using triethylamine as a quaternizing agent. This was achieved using chloromethylation to avoid hazardous conditions. 1H-NMR confirmed the presence of an ammonium moiety in the main polysulfone skeleton. Two A-SPEs with different numbers of cations were obtained, as confirmed by the IEC values, which were similar to those of a commercial anion exchange membrane. In contrast, the KOU value for PSf120 was higher than that of PSf30 and the commercial membrane, indicating that PSf120 exhibits better KOH retention. The developed PSF120 A-SPE demonstrated superior performance and stability compared to commercial A-SPEs in zinc–air batteries (ZABs). Despite the increased internal resistance challenges due to ion transverse transport, the high functionalization of PSF120 enables impedance values comparable to those of commercial A-SPEs. Linear polarization tests demonstrated that PSF120 A-SPE reach higher values of current density (74.22 mA cm^−2^); this was due to PSF120 exhibiting better hydroxide conductivity compared to the A-SPE commercial, demonstrating that hydroxide conductivity could be a limitation for the ZAB operation. In summary, postmortem analysis confirmed the presence of dendritic structures on the zinc electrode; nonetheless, PSf120 A-SPE exhibited an improvement in the reversibility of zinc owing to the higher presence of metallic zinc on the electrode surface. This may be due to the smooth surface of A-SPE PSf120, which does not allow Zn^2+^ to attach to the surface of A-SPE. This is in agreement with the EDS analysis of the air electrode, suggesting that the commercial A-SPE permeates a higher quantity of Zn ions through the separator. Although the formation of carbonates remains a shared challenge, PSF120’s higher electrolyte absorption contributes to its enhanced overall performance. These findings establish PSF120 as a promising material for advancing ZAB technology, offering better efficiency, stability, and reversibility than commercial alternatives.

## Data Availability

The raw data supporting the conclusions of this article will be made available by the authors on request.

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
