# Peer review of "Quaternized Polysulfone as a Solid Polymer Electrolyte Membrane with High Ionic Conductivity for All-Solid-State Zn-Air Batteries"

_membranes, 2025, doi:10.3390/membranes15040102_

Round 1

Reviewer 1 Report

Comments and Suggestions for Authors

The authors synthesized triethylamine-quaternized PSU membranes for use as solid electrolyte membranes in zinc-air batteries. Before further consideration, the following issues should be addressed:

  1. In Figure 2, specific peak positions can be clearly labeled rather than indicating broad spectral bands.
  2. In Table 1, please provide detailed testing information for different membranes, including whether KOH absorption includes water uptake. Additionally, the conductivity testing conditions need to be clarified, and please add theoretical IEC values for comparison with experimental IEC values.
  3. On page 11, the statement, "This suggests that PSf120 enhances zinc reversibility compared to the commercial A-SPE, likely due to a higher distribution of atomic oxygen, indicating that this oxygen is associated with ZnO." lacks direct supporting evidence and need to be further substantiated or revised.
  4. The introduction can be expanded to include discussions on other anion exchange membranes (AEMs) used in different applications to enhance the scope and quality of the paper, such as  Journal of Colloid and Interface Science 2025, 686, 304-317.
Comments on the Quality of English Language

no

Reviewer 2 Report

Comments and Suggestions for Authors
  1. In the introduction section, authors should pay attention in the English language: The following sentence: "The need for renewable energy sources, such solar and wind energy, is critical due to the world economy's rapid development." Should be altered to: "The need for renewable energy sources, such as solar and wind energy, is critical due to the rapid development of the global economy." In order to be more appropriate English language.
  2. The following sentence: "Solid-state electrolytes can eliminate issues associated with liquid electrolyte leakage and evaporation, enhancing the overall safety and environmental friendliness of the batteries." Could be altered to: "Solid-state electrolytes not only eliminate issues associated with liquid electrolyte leakage and evaporation but also enhance the overall safety and environmental sustainability of the batteries." For better clarity.
  3. The introduction shows the qualitative advantages of polymer electrolytes but there is no presentation of numerical comparison in order to have a short quantitative analysis.
  4. Potential challenges of solid-state electrolytes should be added briefly in the text.
  5. Please add an explanation specifying the choice of the benchmark material.
  6. The DC is calculated using NMR integration. It would be informative to briefly mention the role of factor 3 in the equation
  7. You mention that the 3390 cm⁻¹ peak is due to water absorption but do not specify if this is due to polymer hygroscopicity or an artifact of sample handling. If possible, a reference comparing dry and humid spectra would strengthen this claim.
  8. The discussion of TGA results could be clearer regarding why the decomposition temperatures of PSf30 and PSf120 differ (198 °C vs. 208 °C). Is this due to differences in functionalization density, molecular weight, or polymer crosslinking?
  9. The statement "excellent capability for zinc nucleation and redeposition processes at lower overpotentials" is qualitative. If possible, include numerical overpotential values or a comparison with other materials to support this claim.
  10. The phrase "to avoid hazardous conditions" is rather vague. What specific hazards does chloromethylation typically involve? A brief mention of why the modified method is safer would be beneficial.
  11. The higher KOH uptake (KOU) in PSf120 suggests improved electrolyte retention. How does this translate into practical advantages, such as longer battery lifespan or better electrochemical performance? Please provide a reasonable explanation.
  12. In Fig. 5 you mention EIS. This is a Nyquist plot. EIS could be expressed via Nyquist or Bode plots. Please change it.
  13. Authors must alter the axis n Nyquist plot and make the plot tetragonal. For instance, the 100 Ω in the x axis must correspond to 100 Ω in the y axis. Otherwise, the observations derived are misleading.
  1. Moreover, Authors should comment on the difference of the ohmic resistance as obtained from the Nyquist plots. Are these differences related to the induced changes of the materials structure? The potential alterations of the structure are reversible? They should enrich the conversation of the EIS measurements because EIS is a very powerful tool providing plenty of useful information.
  2. It is recommended to add Bode plots also and comment on them.
Comments on the Quality of English Language

Some points require rephrasing for better clarity. Some alterations are highlighted in my comments.
